# Exploring the return-on-investment for scaling screening and psychosocial treatment for women with common perinatal mental health problems in Malawi: Developing a cost-benefit-calculator tool

**Annette Bauer**[1]*, **Martin Knapp**[1], **Jessica Weng**[2], **Dalitso Ndaferankhande**[3], **Edmund Stubbs**[1], **Alain Gregoire**[4], **Genesis Chorwe-Sungani**[3], **Robert C. Stewart**[5]

**1** Care Policy and Evaluation Centre, Department of Health Policy, London School of Economics and Political Science, London, England, United Kingdom, **2** Research Department of Primary Care and Population Health, University College London Medical School, London, England, United Kingdom, **3** Department of Mental Health, Kamuzu University of Health Sciences, Blantyre, Malawi, **4** Global Alliance for Maternal Mental Health, London, United Kingdom, **5** Division of Psychiatry, Centre for Clinical Brain Sciences, University of Edinburgh, Edinburgh, Scotland

* a.bauer@lse.ac.uk

## Abstract

This study sought to develop a user-friendly decision-making tool to explore country-specific estimates for costs and economic consequences of different options for scaling screening and psychosocial interventions for women with common perinatal mental health problems in Malawi. We developed a simple simulation model using a structure and parameter estimates that were established iteratively with experts, based on published trials, international databases and resources, statistical data, best practice guidance and intervention manuals. The model projects annual costs and returns to investment from 2022 to 2026. The study perspective is societal, including health expenditure and productivity losses. Outcomes in the form of health-related quality of life are measured in Disability Adjusted Life Years, which were converted into monetary values. Economic consequences include those that occur in the year in which the intervention takes place. Results suggest that the net benefit is relatively small at the beginning but increases over time as learning effects lead to a higher number of women being identified and receiving (cost-)effective treatment. For a scenario in which screening is first provided by health professionals (such as midwives) and a second screening and the intervention are provided by trained and supervised volunteers to equal proportions in group and individual sessions, as well as in clinic versus community setting, total costs in 2022 amount to US$ 0.66 million and health benefits to US$ 0.36 million. Costs increase to US$ 1.03 million and health benefits to US$ 0.93 million in 2026. Net benefits increase from US$ 35,000 in 2022 to US$ 0.52 million in 2026, and return-on-investment ratios from 1.05 to 1.45. Results from sensitivity analysis suggest that positive net benefit results are highly sensitive to an increase in staff salaries. This study demonstrates the feasibility of developing an

**Data Availability Statement:** All data are available from: https://zenodo.org/records/10533875 DOI: 10.5281/zenodo.10533875.

**Funding:** AB, MK and AG received an award from the Open Society Foundations (grant number OR2019-65006) for conducting the research. The sponsor did not play any role in the study design, data collection and analysis, decision to publish or preparation of the manuscript.

**Competing interests:** The authors have declared that no competing interests exist.

economic decision-making tool that can be used by local policy makers and influencers to inform investments in maternal mental health.

## Introduction

Maternal mental ill-heath during the perinatal period (defined as pregnancy and the first year after delivery) contributes substantially to the global burden of disease [1]. Globally, at least one in five women experience mental health problems during this time, but prevalence rates are much greater in resource-poor settings [2]. In Malawi, an estimated 30% of women experience common mental health problems such as depression, stress or anxiety during the perinatal period [3, 4].

The devastating impacts of perinatal mental illness on maternal mortality and morbidity, as well as on infant mortality and child development, are well established [5–7]. Impacts on children living in poverty in low- and middle-income countries (LMICs) include additional risks of low birth weight, hampered infant growth (linked to reduced breastfeeding and severe malnutrition), severe diarrhoea and low compliance with immunisation schedules [1, 5, 7]. The lifetime costs of untreated perinatal depression and anxiety can be enormous: in previous work, we estimated these costs at US$ 2.8 billion in South Africa and US$ 4.9 billion in Brazil, for example, reflecting the high prevalence and large impacts on health, quality of life and productivity related to the negative consequences for women and children [8, 9].

To address the large impact of perinatal mental health problems, the World Health Organization (WHO) recommends context-appropriate integration of prevention and treatment, in particular psychosocial interventions, into routine maternal healthcare and early child health and development services [1, 10]. Since resources are extremely scarce in most LMICs, approaches for implementing screening and psychosocial interventions (PSIs) have focused on utilising the role of non-specialist community health workers or volunteers integrated into maternal, child health or development services and programmes, in what is known as task-shifting. For example, the WHO endorses scaling up the Thinking Healthy Programme [11], a complex PSI, in which community health workers or volunteers are trained and supervised to deliver cognitive behavioural approaches that address maternal depression in the context of other prevailing risk factors around gender inequity and poverty.

Delivery of the Thinking Healthy Programme (and adapted forms of it) and other PSIs has been trialled in various LMICs, demonstrating improvements in maternal depression [12] as well as infant health-related outcomes, including exclusive breastfeeding, some infant growth and development measures, diarrhoea and immunisation coverage [12, 13]. Implementation evaluations provide evidence on potentially affordable and (cost-)effective ways of delivering PSI at scale, using, for example, task-shifting approaches and digital technologies [11, 14–16]. However, where economic evidence is available, it has been produced in controlled conditions and without considering costs of implementation when delivered at a wider system level or the full range of economic consequences. This limits its use for decision-makers who must make resource allocation decisions with perennially restricted budgets [17, 18].

Since affordability concerns, together with a lack of adequate information about likely costs and benefits linked to the delivery of interventions at scale, have been major barriers to scaling PSIs in LMICs [12, 19, 20], there is a question about how economic analysis can be designed to inform strategic planning and priority-setting. For example, return-on-investment analysis has proven useful in generating an economic case for investment in global (mental) health areas [21, 22]. To be useful to local and regional governments, economic analysis might need

to use country- and context-specific data that reflect local infrastructure and capacities. Considering the paucity of data in many LMICs and the many uncertainties about how to scale the delivery in affordable ways, decision-makers might want to have a tool that allows them to explore the potential impact on costs and returns of different scaling options relevant to the country context.

The aim of this study was to develop a user-friendly decision-making tool that can be used to explore country-specific estimates for costs and returns (i.e., economic consequences or benefits) linked to different options for scaling screening and PSIs for women with common mental health problems (i.e., anxiety and depression) in Malawi. The work was exploratory since we expected many challenges in gathering relevant data, and we needed to make assumptions to overcome these limitations.

## Materials and method

### General approach

Using Microsoft Excel (version 2302) software, we developed a simulation model using a structure and parameter estimates that were established iteratively, based on different information sources and expert views. The model projects current and future populations of women requiring screening and PSIs, number of screenings and PSIs that are delivered, time inputs required to deliver screening and PSIs, costs and economic consequences from 2022 to 2026 at 1-year intervals. We developed the model in the form of a cost-benefit calculator tool that allows decision-makers to select options for key parameters in the delivery of screening and PSIs. Key parameters included as options referred to aspects of delivery for which estimates were uncertain and they had potentially high impacts on the net benefit results. We used a tool developed previously to capture the costs of implementing healthcare innovations [23] as a starting point for designing our tool. We made several adaptations to it. For example, since the original tool only included sheets and sections for calculating costs, we included additional sheets and sections for calculating economic consequences (i.e., returns). The study perspective we took was societal, meaning we included economic consequences not only as they are incurred by government (e.g., healthcare-related expenditure) but also to individuals (e.g., out-of-pocket expenditure). We followed approaches for valuing economic consequences (e.g., productivity losses) used in global mental health economics [22].

In the following sections, we first explain how we gathered the relevant data for the model. Next, we describe the data calculations and assumptions underlying the model. This includes details about what is delivered, how, by whom, as well as the time horizon of the model, and the types of costs and returns considered. Finally, we describe how the tool can be used.

### Information gathering

Iteratively, information was gathered from desk-based searches and from talking to and exchanging emails with experts in the maternal health field to establish a model structure and the parameter values. This included the development of an information request form that presents a list of parameters, parameter values and details about how the values were estimated and the data sources. The information request form was completed iteratively and reflected the knowledge (and knowledge gaps) at different stages of the data-gathering process. Parameters included: effectiveness of PSIs; prevalence rates; population and birth estimates; proportion of women attending antenatal and postnatal visits to health clinics; salaries and reimbursement rates for staff and volunteers delivering PSI; details about screening and PSI delivery (frequency, duration, group size, travelling); details about training and supervision; hospital unit costs; income; inflation; interest and exchange rates; health utility weights; average disease

durations. Where possible, data were gathered specific to Malawi, but wider international evidence was considered where no country-specific data were available, and data were generalisable.

Data were searched from the following sources: published randomised controlled trials and meta-analyses; international databases and resources such as WHO-CHOICE [24], Global Burden of Disease Database [25]; statistical data available from the International Monetary Fund, United Nations Treasury and World Bank; best practice guidance and manuals such as the Global Investment Framework for Women's and Children's Health [26, 27]; Guide for integration of perinatal mental health in maternal and child health services [10]; Thinking Healthy and Problem Management Plus manuals [28, 29].

We consulted two groups of experts: one group included individuals with clinical, research or managerial expertise in funding, managing, delivering, or evaluating screening of common mental health problems and PSIs; the second group included individuals from the Malawi Government, Ministry of Health Reproductive Health Unit and Non-Communicable Disease Committee and Mental Health Unit. The first group of experts included individuals from Kamuzu University of Health Sciences (KUHeS), Partners in Health, Saint John of God Hospitaller, and the University of North Carolina at Chapel Hill. They provided information from research and administrative data systems concerned with implementing and evaluating screening for maternal mental health and the delivery of PSIs (Thinking Healthy Programme, Friendship Bench and Problem Management Plus) in different regions in Malawi, such as: women's attendance rates at health clinics during the perinatal period, frequency and duration of screening and PSIs delivery, proportion of women screening positive for mental health problems, duration of the intervention and lengths of sessions, and size of group (if group delivered). The second group of experts from the Malawi Government provided information on unit costs for hospital use and workforce data, as well as information on how training and supervision might be delivered at scale. Individuals were identified by colleagues of this team based or part-time based in Malawi, which included a psychiatrist specialising in perinatal mental health (RS) and the coordinator of the African Maternal Mental Health Alliance (DN), an organisation concerned with disseminating information and evidence on perinatal mental health to policy makers and influencers, and the wider public.

### Assumptions informing model structure and parameters

**Screening and intervention.**   In line with WHO recommendations, it is assumed that the delivery of PSIs is integrated into maternal and child health care, i.e., health professionals in contact with women (and infants) screen women at the antenatal or postnatal clinic and then refer to PSIs. The model includes the delivery of screening because screening is required as a procedure to identify those women who should be receiving PSIs. Screening is assumed to be delivered in a two-stage process, whereby an initial, very brief (2-minute) screening is done by health professionals (e.g., midwives) in contact with women as part of their antenatal and postnatal care, and this is followed by a second screening (which lasts 10 minutes) provided by the professionals or volunteers trained to deliver the PSI. The decision to include this two-staged screening process in the model was made in consultation with experts because of the very limited capacity of maternity healthcare staff in Malawi to undertake screening. The number of first-stage screenings equals the number of health visits women have during the antenatal and postnatal period, whilst the number of second-stage screenings is an assumed proportion of women screening positive at the first stage. It is assumed that, over time, the identification rate of women with mental health problems increases as practitioners doing the screening are

becoming more confident and competent in that task. We assume that women who screen positive are offered PSIs and that most (95%) accept treatment.

With regards to the PSI delivery, the structure of provision outlined in relevant WHO guidance and manual was followed [30, 31]. As recommended in the guidance, PSIs, such as the Thinking Healthy Programme, are implemented through a task-sharing model, which means that non-specialist (health) staff or volunteers are trained and supervised to deliver the intervention. For this purpose of this exploratory work, we simply assumed that the Thinking Healthy Programme or a similar effective intervention would be scaled. The intervention could be delivered at the clinic or at a community facility and delivered either as a group or on a one-to-one intervention. With regards to the number and duration of sessions, we took estimates as reported in relevant publications of evaluations of PSIs and validated by local experts leading evaluations of PSIs locally [32]. The intervention consists of five sessions: individual sessions last 37.5 minutes, whilst group sessions last 75 minutes. The model does not include costs for procedures linked to referrals and treatment for women with more severe conditions. It is assumed that those women remain unaffected by the implementation of PSIs. Equally, we did not include the provision of antidepressants or other medication for the treatment of common mental health problems in the model, which we assumed would remain unaffected.

**Scaling-up.** We set the scale-up period of 5 years from 2022 to 2026 since the aim was to inform short- to medium-term decision-making. The year 2022 was taken as the start year for calculations. During this period, a scaling-up process is assumed, starting from a zero-provision in 2022 (i.e., no screenings or PSIs are delivered) and increasing linearly to what is considered the maximum possible coverage. The maximum possible coverage is determined by a limited ability to reach out to and identify women with common mental health problems through screening. Whilst there are likely to be restrictions to coverage because of workforce capacities (e.g., not enough midwives to do the screening), for simplicity, the model does not capture those.

**Population.** We estimated the number of women in the antenatal and postnatal periods based on data on birth rates, population, still births and mortality, to which we applied probabilities that women accessing clinics (90% antenatally and 72% postnatally) are screened, identified with mental health problems, and offered treatment. Whilst we assumed the proportion of women accessing clinics to be the same for each year, the number of women offered screening increases linearly from 80% to 100% over the course of the five years, reflecting a learning effect from health professionals, who were assumed to be able to identify more women as it becomes part of their routine. The number of women screened positive at first stage was estimated based on the prevalence of common mental health problems (30%) and sensitivity of screening (80%). The proportion of women screened positive at second stage was estimated based on local data (Table 1).

**Costs.** Costs included in the analysis are those linked to the employment of additional workforce (such as volunteers) required to deliver the screening and PSIs, the costs of training and supervision as well as the costs linked to travelling to provide PSIs.

To calculate the costs of employing additional workforce required to deliver screening and PSIs, the number of fulltime equivalents of professionals or volunteers was calculated based on the total hours required to deliver screening and PSIs each year, divided by number of hours and days that a full-time employed person is working per year. The latter considers legally set working days per year and hours per day. Unit costs for professionals and volunteers were calculated based on their salaries, an overhead rate (15%), and a proportion of direct to indirect time.

To estimate the costs of delivering the training and supervision, we assumed that a train-the-trainer model, also known as a cascading model, is rolled out, which follows the WHO

**Table 1. Parameters, values and data sources that informed the analysis.**

| Parameter | Value | Data source |
|---|---|---|
| *Population* | | |
| 2022 | 20,226,000 | UN World Population Prospects [35] |
| 2023 | 20,809,000 | Ibid |
| 2024 | 21,413,000 | Ibid |
| 2025 | 22,036,000 | Ibid |
| 2026 | 22,679,000 | Ibid |
| Births per 1,000 population, 2020 to 2025 | 35.5 | World Bank population statistics [36] |
| Births per 1,000 population, 2025 to 2030 | 36.0 | Ibid |
| Probability of still births | 2.4% | Study by Makuluni and Stones (2021) [37] |
| Probability of women's death during perinatal period | 0.3% | World Bank mortality statistics [38] |
| Prop. antenatal | 20.1% | Study by Kassebaum et al (2014) [39] |
| Prop. postnatal | 79.9% | Ibid |
| *Live births* | | |
| 2022 | 718,023 | Calculated by dividing total pop by 1,000 and multiplying w births per 1,000 |
| 2023 | 738,720 | Ibid |
| 2024 | 760,162 | Ibid |
| 2025 | 782,278 | Ibid |
| 2026 | 816,444 | Ibid |
| *Proportion of women accessing clinics* | | |
| Antenatal | 90% | Expert views informed by data from Mzuzu & Nsambe clinics in Malawi |
| Postnatal | 72% | Ibid |
| *Proportion of women who are screened (of those visiting clinics)* | | |
| 2022 | 80% | Expert views |
| 2023 | 85% | Ibid |
| 2024 | 90% | Ibid |
| 2025 | 95% | Ibid |
| 2026 | 100% | Ibid |
| Proportion of women screened positive at first stage, ante- and postnatal, 2022 to 2026 | 24% | Based on prevalence rate of 30% from study by Stewart et al (2010) [3] and sensitivity of 80% from study by Chorwe-Sungani & Chipps (2018) [40] |
| *Proportion of women screened positive at second stage, ante- and postnatal* | | |
| 2022 | 12% | Expert views informed by data from Mzuzu & Nsambe clinics in Malawi |
| 2023 | 14% | Ibid |
| 2024 | 17% | Ibid |
| 2025 | 19% | Ibid |
| 2026 | 22% | Ibid |
| Proportion of women accepting treatment | 95% | Expert views |
| *Screening and treatment procedures* | | |
| Duration of first-stage screening, in minutes | 2 | Expert views, based on three-item screening instrument trialled in study in Malawi [40] |
| Duration of second-stage screening, in minutes | 10 | Expert views |
| Number of screenings, antenatal period | 4 | Expert views informed by data from Mzuzu and Nsambe clinics in Malawi |
| Number of screenings, postnatal period | 1.2 | As above |

*(Continued)*

**Table 1.** (Continued)

| Parameter | Value | Data source |
|---|---|---|
| Number of treatment sessions | 5 | Study of PM+ programme by Dawson et al (2015) [32] |
| Duration of individual treatment session, in minutes | 37.5 | Expert views; reflects midpoint between 30 to 34 minutes of average session durations observed for Thinking Healthy and Friendship Bench programmes in Malawi |
| Duration of group—based treatment session, in minutes | 75 | Expert views; reflects midpoint of 60 and 90 minutes observed for PM+ programme in Malawi |
| Group size | 6 | Reflects midpoint of 4 and 8 7observed for with PM + programme in Malawi |
| *Work terms and conditions* | | |
| Ratio of direct/ indirect working time for professionals, midwives | 80% | WHO-Choice and A Global Investment Framework for Women's and Children's Health (WHO 2013) 26 |
| Ratio of direct/ indirect working time for professionals, counsellors | 80% | Ibid |
| Working hours per day, midwives | 8 | Ibid |
| Working hours per day, counsellors | 6 | Ibid |
| Days worked per year, midwives | 220 | Ibid |
| Days worked per year, counsellors | 220 | Ibid |
| Overhead rate, health professionals | 15% | Derived from study by Chisholm et al (2016) [22] |
| Overhead rate, volunteers | 15% | As above |
| Training and supervision | | |
| *Hours of time spent on training led by master trainer* | | |
| Classroom training | 40 | Derived from study by Msisuka et al (2011) [33] |
| Field training | 20 | Ibid |
| Additional training in supervision | 16 | Ibid |
| *Hours of time spent on training led by trained trainer* | | |
| Classroom training | 88 | Derived from WHO manual for PM+ [29] and study by Msisuka et al (2011) [33]; refers to average between training for individual and group-based sessions |
| Field training | 17.5 | Ibid |
| Refresher training provided annually 2023 to 2025 | 16 | Ibid |
| Hours of time spent on 1-2-1 supervision of trainee counsellors, per year | 44 | Ibid |
| Number of course participants | | |
| • course led by master trainer | 12.5 | Ibid |
| • course led by trainer | 5 | Ibid |
| Fee per hour of master trainer (clinical psychologist), in MWK | 2,273 | Expert views; calculated based on monthly salary of MWK 400,000, 22 days worked per month and 8 hours per day |
| Fee per hour of trainer (professional assistant counsellor), in MWK | 1,364 | Expert views; calculated based on monthly salary MWK 240,000, 22 days worked per month and 8 hours per day |
| Travel costs per journey, in MWK | 300 | Derived from study by Zumazuma (2020) [41] |
| *Effectiveness* | | |
| Reduction in mental illness, ante- and postnatal period | 6% | Derived from study by Sikander et al (2019) [11]; refers to remission at 3 months in intervention vs. control group of 50% vs. 44% |
| Reduction in productivity loss | | |
| • additional days able to work, antenatal period | 0.765 | Derived from study by Sikander et al (2019) [11]; refers to average number of days unable to work in last month in intervention vs. control group of 1.18 vs. 1.26, which is multiplied by 9 months |

(*Continued*)

**Table 1.** (Continued)

| Parameter | Value | Data source |
|---|---|---|
| • additional days able to work, postnatal period | 1.02 | Derived from study by Sikander et al (2019) [11]; refers to average number of days as described above, which is multiplied by 12 months |
| Reduction in infant diarrhoea episodes, postnatal period | 11% | Derived from study by Rahman et al (2008) [34]; refers to proportion of infants with diarrhoea episodes at 12 months in intervention vs. control group of 32% vs. 43% |
| Reduction in stunting | 5% | Derived from study by Rahman et al (2008) [34]; refers to proportion of infants with stunting at 12 months in intervention vs. control group: 18% vs. 23% |
| *Disability weights* | | |
| • Moderate major depressive disorder | 0.40 | Global Burden of Disease study by Burstein et al (2015) [25] |
| • Moderate diarrhoea | 0.19 | Ibid |
| *Average durations* | | |
| • Moderate major depressive disorder, in years | 0.5 | Estimated average from studies by Cox et al (1993) [42], Spijker et al (2002) [43] and ten Have et al (2017) [44] |
| • Moderate diarrhoea, in days | 1.44 | Study by Lamberti et al (2012) [45] |
| *Costs, in MWK* | | |
| Treatment of infant diarrhoea | 32,359 | Derived from study by Hendrix et al (2017) [46]; refers to weighted average of rural and urban inpatient and outpatient costs for treating an episode of acute childhood gastroenteritis in Malawi, updated to 2021 using Consumer Price Index |
| Income of a woman, per day | 455 | Castel et al (2010) [47]; refers to daily income by female employees uprated from 2005 to 2021 prices using GDP data |
| *Gross Domestic Product, per capita* | | |
| 2022 | 544 | International Monetary Fund data [48] |
| 2023 | 521 | Ibid |
| 2024 | 509 | Ibid |
| 2025 | 505 | Ibid |
| 2026 | 509 | Ibid |
| *Inflation rate based on consumer price index* | | |
| 2022 | 9% | International Monetary Fund data [49] |
| 2023 | 7% | Ibid |
| 2024 | 6% | Ibid |
| 2025 | 5% | Ibid |
| 2026 | 5% | Ibid |

manual for Problem Management Plus and is described in implementation studies [33]. In the train-the-trainer model, a lead (or 'master') trainer, who is a clinical psychologist, provides classroom teaching, field training and supervision. Individuals trained by the 'master' trainer then provide training and supervision to other trainees. Costs of training refer to the hours spent by the master trainer and the newly trained trainers on delivering the training, as well as hours spent by trainees attending the master training and the training of the newly trained trainers. Costs were first calculated per course and then multiplied by the number of full-time equivalents of staff who need to be employed (calculated as described above) based on a fixed number of trainees per course.

Costs of supervision, calculated per trainee, refer to the hours spent by the trained trainers for providing supervision and the hours spent by the trainees for receiving supervision. Costs

per trainee were then multiplied by the number of full-time equivalents required to provide the training per year (since supervision is assumed to be required on an ongoing basis).

Costs for travel refer to those linked to travelling required by women participating in PSIs (if the intervention is provided at the health clinic) or travelling by professionals or volunteers (if the PSI is provided in the community). It is assumed that for PSIs delivered in the community travel incurs to the professional or volunteer providing the treatment, whilst for the PSI delivered in health clinics travel time is incurred by women (but not for the professionals or volunteers providing the intervention). Costs were first calculated per woman based on number of sessions, group size (for group-based interventions), and travel cost per journey, and then multiplied by the number of women receiving PSIs. We did not include travel costs linked to screenings since those visits would happen anyway as part of regular maternal healthcare.

**Economic consequences.** Following a conservative approach, economic consequences included in the model refer to a short-term perspective of one year, which means we only included the consequences that occur in the same year that the intervention is delivered. Economic consequences include reductions in healthcare expenditure (linked to a reduction in hospital episodes for the treatment of infant diarrhoea), in women's productivity losses and in health-related quality of life losses (for women and infants).

We calculated the healthcare savings linked to a reduction in hospital episodes for the treatment of infant diarrhoea for mothers receiving PSIs by calculating the difference in hospital episodes between intervention and control groups found in published trials and attaching the unit cost for the treatment of an episode of acute infant gastroenteritis. Unit costs are a weighted average of rural and urban inpatient and outpatient costs, with weights reflecting proportions treated in the different settings.

In line with analysis approaches employed by the World Health Organization [22], we valued outcomes of PSIs in terms of disability-adjusted life years (DALYs) prevented (for mothers and infants) as well as productivity gained (for mothers).

We calculated gains in women's health-related quality of life linked to the additional women who recovered from depression because of PSI by taking the difference in remissions from depression in intervention and control groups from trial data [11] and assigning an average duration of illness and a disability weight for moderate depression. This provides us with total DALYs averted. The DALYs-averted-per-woman estimate is then applied to the number of women receiving PSIs (estimated as described in the Population subsection above). This calculation does not include any potential reduction in excess risk of premature mortality. We calculated infants' improved quality of life linked to a reduction in children experiencing diarrhoea because of the PSI in a similar way. We multiplied the difference in diarrhoea episodes between the intervention and control group as identified by trials [34] with the average disease duration and disability weight for moderate diarrhoea. This is equivalent to the estimated DALYs averted because infants of women receiving PSIs are less likely to experience diarrhoea episodes. The DALYs-averted-per-infant estimate was multiplied by the number of women receiving PSIs.

To calculate productivity gains for women accessing PSIs, we first calculated the additional days women can work during the perinatal period based on the additional days women in the intervention group are able to work compared to a control group, as identified in a large trial [11] and multiplied this by an average hourly income for women and average number of hours worked per day. The average additional days per woman were then multiplied by average daily income of female employees in Malawi, and this amount was applied to women receiving PSI in the model.

**Other consequences.**   Since children of women receiving PSIs are less likely to be stunted [34], we included this outcome in the analysis. Whilst stunting has been associated with many adverse long-term outcomes, there is a lack of evidence concerning immediate economic consequences. We calculated the reduction in stunting by taking the difference in stunting in the first year between intervention and control group from trial data [34].

All parameters, values used for the analysis and their data sources are presented in Table 1.

## Tool description

**Design and structure.**   The cost-benefit-calculator tool is a Microsoft Excel (version 2331) document and includes simple Visual Basics for Applications coding. It is structured into different sections (worksheets) which the user can navigate by clicking on fields with headings. The structure of the tool is designed for two types of users: (1) decision-makers who can use the tool to explore the impact of changing options provided for selected key parameters on the cost-benefit results; and (2) technical persons who are familiar with the data and assumptions that inform the results and can therefore change any of the parameter estimates. The headings of the worksheets are titled as follows: home or introduction (which provides basic instructions for how to use the tool); options (where the user can enter their choices and see the immediate impact on the results which are presented in graphical and numerical form); a detailed results section (which includes all outputs that inform the results, such as number of women screened and treated, number of first- and second-stage screenings and PSIs delivered, number of full-time equivalents required to deliver screenings and PSIs; costs for the population by types and in total; outcomes for the population by types and in total). In addition, there is a worksheet 'administration', from which the user (technical person) can navigate to the different worksheets that present the parameters and values underlying the cost-benefit results on: population, intervention, workforce, training and supervision, travel, effectiveness, and economic consequences.

**Options.**   Choosing from a given range of values, users (decision-makers) can change the values of the following parameters: salaries of professionals conducting the first-stage screening; salaries of professionals or volunteers conducting the second-stage screening and delivering the PSIs; setting (clinic versus community-based); group versus individual sessions. Options were provided because there was either substantial variation or no clear view among experts on their values. For example, there was substantial uncertainty as to which professional group or volunteers should be providing the second-stage screening and PSI (and thus which salaries or reimbursement rate would need to be considered). Options are provided in a drop-down menu whereby the user can select a value among a limited number of options and see the cost-benefits results linked to the selected value (or combination of values). For the salaries, values are provided in numbers, whilst for the options concerning setting and format, options are given in proportions (0%, 25%, 50%, 75% and 100%). All other parameters in other parts of the Excel document can also be changed, but this should only be done by the technical person familiar with the tool.

**Key outputs.**   The key outputs from the analysis are year-on-year estimates of (1) the costs of conducting screenings and delivering PSIs, training, supervision, and travel, and (2) economic consequences as result of treating women with PSIs including savings in healthcare expenditure, gains in productivity and reductions in DALYs. Inflation and discount rates were applied to total costs and total benefits to generate present values in 2022 US$. Findings are presented in net benefits (equal to total costs minus total benefits) and return-on-investment ratios (equal to benefits divided by costs). In the tool, results are presented in both graphical and numerical form.

## Results

Since the results depend on the selected inputs, results are presented for a scenario with a randomly chosen combination of selected inputs. For this scenario, it is assumed that first-stage screenings are provided by health professionals (e.g., midwives), who earn US$ 235 per month, and that second-stage screenings and PSIs are provided by volunteers, who receive a small payment of US$ 40 per month. It is also assumed that an equal proportion of women receive PSIs in group versus individual session format and at clinic versus community settings. Results are presented in Fig 1 and Table 2. Fig 1 presents a visualisation of the Excel sheets showing the user surface with options and results. (A link to the repository record with access to the tool is provided in the S1 File). Table 2 shows the more detailed results provided by the tool, including the total number of screenings and PSIs delivered each year nationally, the number of health professionals and volunteers required to deliver those (in full-time equivalents), as well as costs and benefits (by categories).

For example, in 2022, just over 2.2 million first-stage and just over half a million second-stage screenings would be conducted, with numbers increasing to 3.2 million and 0.76 million in 2026. A total of 53 health professionals and 120 volunteers would be required in 2022 and the figure would increase to 75 and 213 in 2026. Total estimated costs are US$ 0.66 million, which include the costs to government for employing, training and supervising staff, and paying travel reimbursements. Regarding health benefits, it is estimated that a total of 1,123 DALYs can be averted in 2022, which is equivalent to a monetary value of US$ 0.36 million. This figure increases to 2,881 DALYs and US$ 0.93 million in 2026. The vast majority of DALYs averted relate to health benefits to mothers and only a very small proportion to those for infants. Reduction in healthcare expenditure linked to prevented cases of diarrhoea are US$ 0.26 million in 2022 and US$ 0.65 million in 2026. Productivity gains are US$ 0.13 million in 2022 (reflecting 164,610 additional days worked) and US$ 0.42 million in 2026 (reflecting 653,927 additional days worked). Net present values using a discount rate of 3% are under US$ 35,000 in 2022 and increase to US$ 0.52 million in 2026. The return-on-investment ratio in 2022 is 1.05 in 2022 and increases to 1.45 in 2026. In addition, the reduced number of children who would be growing up stunted is estimated to be 2,105 in 2022 and 5,385 in 2026.

Findings from our analysis suggest that the net present economic benefit is relatively small initially but increases over time as assumed learning effects lead to a higher number of women being identified and receiving (cost-)effective treatment. Positive net benefits are highly sensitive to an increase in staff salaries. For example, when we assumed that volunteers (or staff) delivering PSIs would be paid about US$ 50 per month, the net benefit would become negative for the first year and the return-on-investment ratio in the final year would only be 1.21. If volunteers or staff are paid US$ 200 per month, net benefits are negative across the five-year period. Changing how treatment is delivered (i.e., group versus individual or clinic versus community) only affects net benefits marginally.

## Discussion

In this paper we describe an exploratory economic analysis conducted to demonstrate the feasibility of developing a cost-benefit calculator tool designed to help decision-makers systematically examine the projected costs and benefits, as well as necessary resource requirements, of scaling-up screening and PSIs for women with common mental health problems in Malawi. This kind of tool provides a collection of relevant information for planning future implementation and highlights changes in costs and benefits over the years, under certain delivery assumptions. It can be used to make a potential investment case for scaling-up screening and treatment for common perinatal mental health problems. It also provides information about

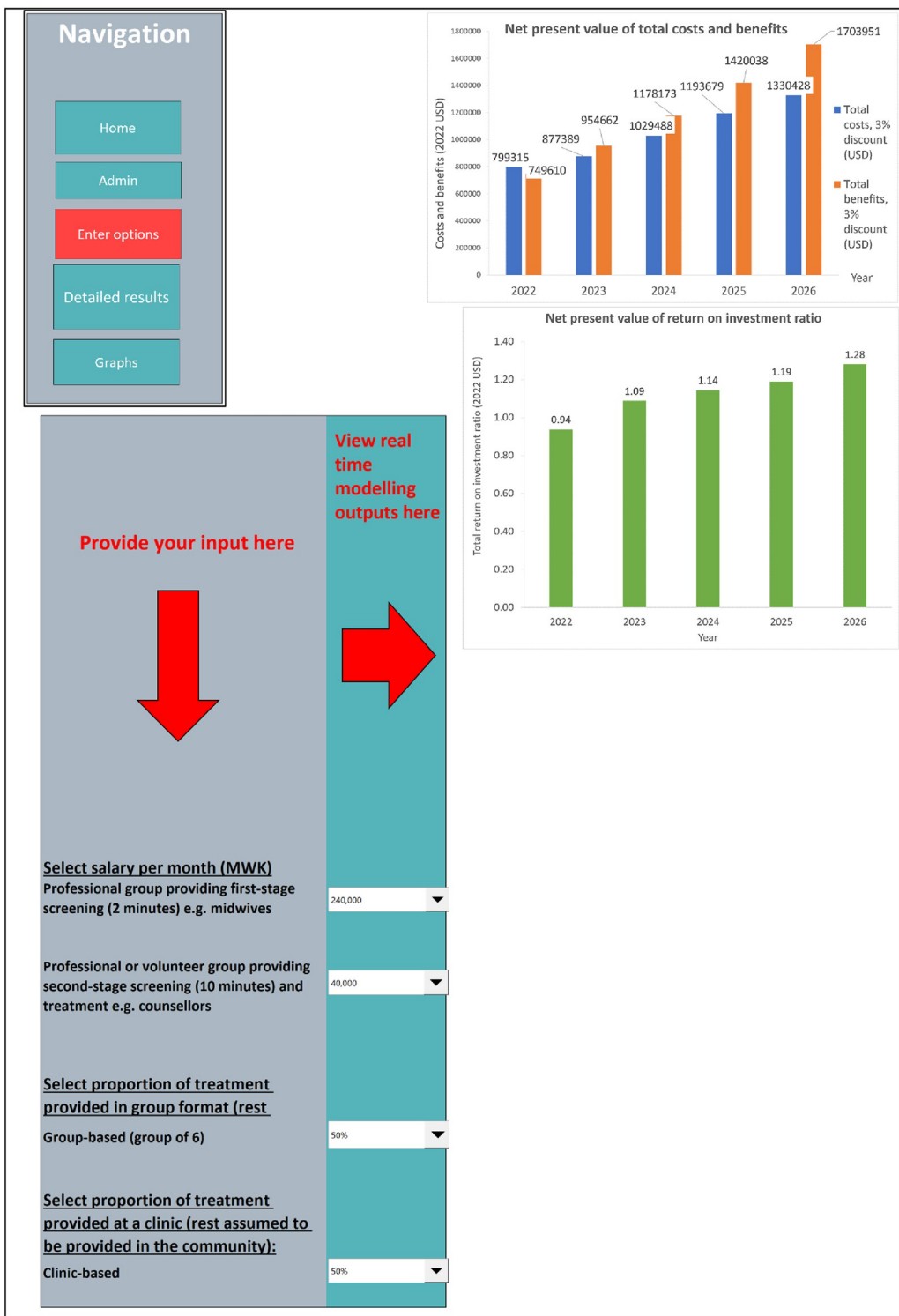

**Fig 1. Results from return-on-investment analysis (for the chosen scenario), as presented in cost-benefit calculator tool.**

**Table 2. Results from return-on-investment analysis (for the chosen scenario), summarised.**

| | 2022 | 2023 | 2024 | 2025 | 2026 |
|---|---|---|---|---|---|
| **Procedures, workforce** | | | | | |
| No. screenings (first-stage) | 2,230,858 | 2,438,608 | 2,657,002 | 2,886,212 | 3,170,808 |
| No. screenings (second-stage) | 535,406 | 585,266 | 637,681 | 692,691 | 760,994 |
| No. women treated | 92,219 | 120,968 | 153,768 | 190,895 | 235,933 |
| Total days required for first-stage screening | 9,295 | 10,161 | 11,071 | 12,026 | 13,212 |
| Total days required for second-stage screening and treatment | 21,276 | 24,658 | 28,392 | 32,498 | 37,523 |
| Total number of health professionals required | 53 | 58 | 63 | 68 | 75 |
| Total number of volunteers required | 121 | 140 | 161 | 185 | 213 |
| **Costs** | | | | | |
| Costs staff time, in MWK | 260,356,092 | 287,346,456 | 316,070,578 | 346,585,088 | 384,328,742 |
| Costs training, in MWK | 32,951,846 | 866,084 | 997,229 | 1,141,459 | 1,317,955 |
| Costs supervision, in MWK | 7,857,780 | 9,106,620 | 10,485,567 | 12,002,106 | 13,857,910 |
| Costs travel, in MWK | 71,597,676 | 93,918,321 | 119,384,240 | 148,209,243 | 183,176,388 |
| Total costs (aggregated), in MWK | 372,763,394 | 391,237,481 | 446,937,613 | 507,937,896 | 582,680,995 |
| **Total costs (aggregated), in USD** | **656,274** | **688,798** | **786,862** | **894,257** | **1,025,847** |
| **Benefits** | | | | | |
| Episodes of perinatal mental illness prevented (mothers) | 5,670 | 7,438 | 9,454 | 11,737 | 14,506 |
| DALYs linked to perinatal mental illness averted (mothers) | 1,123 | 1,473 | 1,872 | 2,324 | 2,872 |
| Episodes of diarrhoea prevented (children) | 4,487 | 5,885 | 7,481 | 9,287 | 11,479 |
| DALYs linked to diarrhoea averted (children) | 3.33 | 4.37 | 5.55 | 6.89 | 8.51 |
| Total DALYs averted | 1,126 | 1,477 | 1,877 | 2,331 | 2,881 |
| **Total health benefits, in USD** | **362,002** | **474,856** | **603,614** | **749,355** | **926,150** |
| **Healthcare savings linked to reduction in diarrhoea (children), in USD** | **255,599** | **335,282** | **426,193** | **529,097** | **653,927** |
| Additional number of days of work (mothers) | 164,610 | 215,927 | 274,476 | 340,748 | 421,140 |
| **Productivity gains (mothers), in USD** | **132,009** | **173,163** | **220,116** | **273,263** | **337,734** |
| Reduced number of stunting (children) | 2,105 | 2,761 | 3,510 | 4,357 | 5,385 |
| Net benefit (= health benefit + productivity gains + healthcare savings—total costs), in USD, not discounted | 93,336 | 294,503 | 463,061 | 657,457 | 891,965 |
| **Net present value, discounted at 3%, in USD** | **34,272** | **175,444** | **264,684** | **361,713** | **525,254** |
| **Return-on-investment ratio** | **1.05** | **1.23** | **1.29** | **1.34** | **1.45** |

variations in return-on-investment depending on different scaling scenarios. The analysis was developed collaboratively with experts, which is increasingly recommended for economic evaluations to incorporate real-world implementation conditions pertaining, for example, to staff capacity, geographical circumstances and socio-cultural norms [50, 51].

The analysis highlighted various challenges and limitations that would need to be addressed in future economic evaluations of this kind. First, the limited availability of both research evidence and routinely collected data (e.g., on current screening rates, currently employed workforce) meant that we needed to make various assumptions about the model structure as well as the parameter estimates. Second, this study only had limited resources to bring together experts from across different types of organisations (e.g., government, non-government organisation, universities) or staff groups (e.g., mental health, maternity) to discuss and resolve areas of uncertainty or disagreement. It was challenging to gather relevant information from experts who, generally, have extremely busy time schedules which can change at short notice due to emergencies, disruptions or other reasons. The fact that we were able to gather enough data for this analysis despite having no funds to pay for their time suggests that experts were

interested in and supportive of this area of work. Poor internet connections made it difficult to set up or hold online meetings effectively. Third, the analysis only includes short-term consequences, whilst many of the expected benefits or costs consequences are likely to be long-term. For example, our analysis suggests that up to 2,000 cases of childhood stunting per year can be potentially averted if women receive PSI. This would present sizable economic benefits over the longer term: it has been estimated that stunting can lead to up to 10% loss in lifetime earnings [52], and a substantial loss of 1% to 11% in gross domestic product [53, 54].

Despite these limitations, our research demonstrates the potential usefulness of this type of analysis and tool in informing scaling decisions by exploring the impact of changing assumptions about parameters on economic value. The anticipated cost of implementation has been identified as the main barrier to scaling [50, 51]. Future research would need to employ more comprehensive consultation processes, which also requires appropriate research resources, for example reimbursement of participants' time or at least reimbursement of expenses incurred in participation in meetings. This should include a detailed planning of different scaling scenarios and setting out resource requirements to implement them. This process also needs to be designed to achieve buy-in from experts and other stakeholders. This might include exploring their motivations for engaging in the processes, building consultation processes around their preferences and abilities, and building research capacity.

Delphi processes can be particularly helpful in gaining agreement on certain questions about model structure (e.g., whether to include a first- and second-stage screening process) and parameter estimates (e.g., how to adapt effectiveness data under different delivery assumptions) [55]. Especially as data for this kind of analysis are, by definition, unknown (i.e., it is unclear how data that have been established under trial conditions translate under real-world conditions), the role of experts in making informed assumptions is essential. Since interventions in trials are often considered unaffordable for scaling at national level, one application of this kind of economic tool might be to inform decisions about the design of large trials—i.e., how the intervention might need to be delivered to achieve a positive return on investment. The input variables needed by the tool should also guide researchers in identifying important variables to measure in such implementation research. This could provide funders with information to fund research that is more likely to lead to sustainable adoption.

In conclusion, our analysis provides useful proof-of-concept for conducting return-on-investment analysis for scaling screening and psychosocial treatment for women's mental health during the perinatal period. We believe that the research is an important step in the development of a methodology and tool that can be applied in other countries using country-specific inputs to inform resource allocation decisions in maternal mental healthcare. Future analysis could make greater use of machine learning to systematically explore associations between variables and identify factors driving costs and consequences.

## Supporting information

**S1 File. Repository link for access to the data.**
(DOCX)

## Acknowledgments

We would like to acknowledge the contributions to this study from the following individuals and organisations: Dr Michael Udedi (NCDs & Mental Health Unit, Malawi Ministry of Health), Dr Fannie Kachale (Reproductive Health Services, Malawi Ministry of Health), Dr Mwawi Ng'oma (Saint John of God Hospitaller Services, Lilongwe), Dr Bradley Gaynes

(Department of Psychiatry, University of North Carolina School of Medicine), Professor Mina Hosseinipour (UNC Project Malawi and University of North Carolina School of Medicine), Dr Brian Pence (Department of Epidemiology, Gillings School of Global Health, University of North Carolina School of Medicine), Dr Ryan McBain (Brigham and Women's Hospital, Harvard Medical School), Dr Todd Ruderman (Partners in Health, Abwenzi Pa Za Umoyo).

## Author Contributions

**Conceptualization:** Annette Bauer, Martin Knapp, Robert C. Stewart.

**Data curation:** Annette Bauer, Jessica Weng, Dalitso Ndaferankhande, Genesis Chorwe-Sungani.

**Formal analysis:** Annette Bauer, Jessica Weng.

**Funding acquisition:** Annette Bauer, Martin Knapp, Alain Gregoire.

**Investigation:** Annette Bauer, Jessica Weng, Dalitso Ndaferankhande, Alain Gregoire.

**Methodology:** Annette Bauer, Martin Knapp, Jessica Weng, Robert C. Stewart.

**Project administration:** Annette Bauer, Dalitso Ndaferankhande, Alain Gregoire.

**Resources:** Annette Bauer, Dalitso Ndaferankhande, Alain Gregoire.

**Software:** Annette Bauer, Edmund Stubbs.

**Supervision:** Martin Knapp, Robert C. Stewart.

**Validation:** Annette Bauer, Alain Gregoire, Genesis Chorwe-Sungani.

**Visualization:** Annette Bauer, Edmund Stubbs.

**Writing – original draft:** Annette Bauer.

**Writing – review & editing:** Annette Bauer, Martin Knapp, Jessica Weng, Dalitso Ndaferankhande, Edmund Stubbs, Alain Gregoire, Genesis Chorwe-Sungani, Robert C. Stewart.

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
